# Microglial Turnover in Ageing-Related Neurodegeneration: Therapeutic Avenue to Intervene in Disease Progression

**DOI:** 10.3390/cells10010150

**Published:** 2021-01-14

**Authors:** Shofiul Azam, Md. Ezazul Haque, In-Su Kim, Dong-Kug Choi

**Affiliations:** 1Department of Applied Life Science & Integrated Bioscience, Graduate School, BK21 Program, Konkuk University, Chungju 27478, Korea; shofiul_azam@hotmail.com (S.A.); mdezazulhaque@yahoo.com (M.E.H.); 2Department of Integrated Bioscience & Biotechnology, College of Biomedical and Health Science, Research Institute of Inflammatory Disease (RID), Konkuk University, Chungju 27478, Korea; kis5497@hanmail.net

**Keywords:** microglia, neurodegeneration, neuroinflammation, macrophages, homeostasis

## Abstract

Microglia are brain-dwelling macrophages and major parts of the neuroimmune system that broadly contribute to brain development, homeostasis, ageing and injury repair in the central nervous system (CNS). Apart from other brain macrophages, they have the ability to constantly sense changes in the brain’s microenvironment, functioning as housekeepers for neuronal well-being and providing neuroprotection in normal physiology. Microglia use a set of genes for these functions that involve proinflammatory cytokines. In response to specific stimuli, they release these proinflammatory cytokines, which can damage and kill neurons via neuroinflammation. However, alterations in microglial functioning are a common pathophysiology in age-related neurodegenerative diseases, such as Alzheimer’s, Parkinson’s, Huntington’s and prion diseases, as well as amyotrophic lateral sclerosis, frontotemporal dementia and chronic traumatic encephalopathy. When their sentinel or housekeeping functions are severely disrupted, they aggravate neuropathological conditions by overstimulating their defensive function and through neuroinflammation. Several pathways are involved in microglial functioning, including the *Trem2*, *Cx3cr1* and progranulin pathways, which keep the microglial inflammatory response under control and promote clearance of injurious stimuli. Over time, an imbalance in this system leads to protective microglia becoming detrimental, initiating or exacerbating neurodegeneration. Correcting such imbalances might be a potential mode of therapeutic intervention in neurodegenerative diseases.

## 1. Introduction

Microglia are brain-dwelling parenchymal macrophages [1] that are distinct from other brain-dwelling non-parenchymal macrophagic populations and tissue macrophages [2,3]. Under healthy and normal conditions, microglia are self-maintained with no considerable contribution of peripheral myeloid cells [4]. In addition, ~5% of the total cells of the neocortex are normally microglial [5]. These macrophages have a uniform distribution throughout the central nervous system (CNS) parenchyma, with a population of approximately 10% of the total cells in the CNS. In addition, they can form a cellular grid with their ramified and highly motile process [6]. During the developmental phase of the brain, microglia help shape neural circuits by modulating the strength of synaptic transmissions and sculpting neuronal synapses. When sensing any CNS injury, they become phagocytic and eliminate microbes, cell debris, protein aggregates or any form of CNS insult. However, microglial activation has been reported during several neurological conditions [7,8], in which they have evidently been playing both beneficial and detrimental roles, depending on disease progression.

The correct functioning of microglia is necessary for neural circuit remodelling and synaptic function, but their involvement in the pathogenesis of neurological diseases raises the debate of whether microglial activation is beneficial or detrimental. In response to CNS insults, microglia proliferate rapidly [9], but their proliferation and turnover rates under homeostatic conditions have not been well documented. Filling this research gap could differentiate them from disease-causing agents [10]. Several studies have shown interesting but conflicting results, either very low or very high microglial turnover in mice and the post-mortem brain [11,12,13], although the validity of these techniques has been arguable and limited [14,15]. However, it is clear that understanding the regulation of microglial populations in the brain is crucial to understanding their functioning at different phases of age and age-related diseases. In one case, the mouse model of AD showed that early priming of individual microglia can induce long-lasting functional changes in life, whereas microglial senescence may have a role in age-related neurodegeneration [10]. For example, stimulation of the brain’s immune system during development can induce long-lasting changes in microglial immune responses [16,17], and ageing and senescence of microglia contribute to neurodegenerative disorders [18].

However, changes in microglial homeostasis and functioning during age and age-related neurodegenerative diseases could be due to alterations in the brain’s microenvironment or to the longevity of microglia, possibilities that remain unclear. Therefore, this article will discuss in detail microglia’s physiological role and changes due to exposure to different insults in individual age-related neurodegeneration.

## 2. Molecular and Functional Background of Microglia

Microglia are the resident immune cells of the brain and cover almost 5–12% of CNS cells [19]. In addition, microglia are involved in the homeostasis of host defence against pathogens and consequent neurological disorders [20,21]. These cells are mesenchymal, originating in the yolk sac, and do not require hematopoietic stem cells for renewal [13,22]. Their survival and maintenance depend on cytokines, including CSF1 and interleukin (IL)-34 [23], and on transcription factors such as IRF8 [22]. However, microglia can be simply defined as innate immune cells of the CNS that originate from myeloid cells and express several genes, including *Cx3cr1*, *CD11b*, *Iba1* and *F4/80* [21]. Depending on comprehensive knowledge of microglial gene expression and relevant functions [21,24], this study attempted to determine microglial functions in accordance with their gene expressions. There are three basic functions of microglia—sensing their environment, maintaining physiological homeostasis and protecting against self-modified and exogenous injurious agents. Furthermore, these normal functions are important regulators from a human being’s embryonic stages through to old age.

Sensing is the primary requirement for microglia to function in housekeeping and defending their host from injury (Figure 1). Microglia form a network that spans throughout the CNS. Their thin processes are dynamic and in constant motion, allowing them to scan the area surrounding their cell body every few hours and rapidly polarise toward focal injury [8,19]. This network includes approximately 100 or more genes that consistently scan surrounding cell bodies and sense any changes in their microenvironment [24,25,26]. The sensome mRNA is expressed uniformly in microglia of different regions of the brain, indicating that all microglia are capable of sensing functions.

The second and most important part of microglial function is physiological homeostasis. This function includes synaptic remodelling such as CNS development and maintenance, neurodegeneration [27,28], phagocytosis of dead or malfunctioning neuronal cells or cell debris [29,30], or myelin homeostasis [31] (Figure 2). In addition, microglia activate several inflammatory pathways that cause neuroinflammation and possibly neurodegeneration. In this process, microglia interact with astrocytes, and their interaction is also important in regulating homeostasis. Several chemokine and chemoattractant housekeeping genes are involved in phagocytosis (*Trem2*), synaptic pruning and remodelling (*C1q* and *Cx3cr1*) [24], and anomalies in these housekeeping genes may lead to neurodegeneration.

As carriers of innate immunity, microglia defend their host against pathogens (Figure 1), injurious proteins including Aβ, aggregated α-synuclein, mutant huntingtin (*mHtt*), mutant prion (mPrP^Sc^) and oxidised superoxide dismutase (SOD). Microglia activate several receptors to incite host defences, such as expressing Fc receptors, Toll-like receptors (TLRs), and several antimicrobial peptides including *Camp* and *Ngp* [24]. Therefore, microglia approach the neuroinflammatory threshold by producing peripheral inflammatory cytokines such as TNF-α and IL-1β [33,34]. In addition, this process involves chemokines (e.g., Ccl2) that recruit additional cells and work together to clear pathogens and normalise the brain in a homeostatic condition [35]. However, consistent microglia-induced neuroinflammation leads to neurotoxicity and, eventually, neurodegeneration. In contrast to the anomalies in this microglial functioning, healthy neural microglia are always actively functioning through sensing, housekeeping and protecting their host.

## 3. Microglia and Ageing

Ageing is inevitable, and several structural and functional changes occur in the brain with advanced ageing. For instance, the brain loses a total mass of about 2 to 3% per decade after the age of 50. This loss of mass with age specifically affects the volume of grey and white matter in the prefrontal, parietal and temporal areas [36,37,38]. Therefore, an individual gradually loses complex learning abilities and declines in cognitive function [38,39]. Several cellular-level changes also occur in ageing brains, such as genomic instability, shortening of telomeres and activation of tumour suppressor genes, protein mutation and accumulation, oxidative stress, reduced autophagy and mild to chronic inflammation. It is imperative to maintain the balance between pro- and anti-inflammatory cytokines. However, a brain undergoing advanced ageing shows an imbalance between these cytokine levels in response to chronic exposure to physical, chemical or biological agents, such as ionic radiation, pollutants and pathogens [40,41]. Studies have shown that chronic exposure to endogenous or exogenous pathogens decreases the anti-inflammatory cytokine IL-10 [42]. In contrast, such exposure also increases inflammatory cytokines such as TNF-α and IL-1β in the CNS [43] and IL-6 in plasma [44]. Additionally, increased systemic inflammation causes neuronal cell death and an imbalance between clearance and production of ROS, severely damaging synaptic plasticity as well (Figure 2 and Figure 3). Many of these alterations in ageing brains include impairment in basal autophagy that begins with cellular stress. Ageing human brain analysis has shown a reduction in autophagy genes, including *Atg5*, *Atg7* and *Becn1* [45], and similar downregulation in Atg-proteins has been evident in ageing mouse brains. In contrast, ageing has been found to upregulate mTORC1 [45,46] and accelerated mTOR reduces macroautophagy and promotes aggregated protein and metabolic disturbances during ageing. These data have been supported by the rapid development of neurodegeneration in *Atg5*- [47] and *Atg7*- depleted [48] mice. Microglia, as the first line of host defence, selectively activate autophagy to entrap threatening molecules into autophagosomes and clear them via autophagic degradation. For example, microglial TLR4-induced activation of nuclear factor κB (NF-κB) upregulates p62/SQSTM1 signalling, which degrades misfolded α-syn proteins via autophagy and protects against midbrain dopaminergic neuronal loss [49]. Furthermore, ageing-mediated alteration in microglial functions disrupts microglial regulation of autophagy and promotes neuronal loss.

Ageing has been recognised as a major risk factor for many neurodegenerative disorders. Advanced ageing includes several hallmarks indicating risks of developing neurodegenerative diseases such as Alzheimer’s (AD), Parkinson’s (PD), Huntington (HD) and frontotemporal lobar (FTD) disease. Furthermore, microglial cells change with ageing, which is one of the major risk factors for age-related development of neurodegeneration. Although neurodegenerative diseases are multifactorial conditions, and their complexity is not yet well understood, there has been scientific agreement on the degenerative diseases and age-related changes they can cause in the neural microenvironment.

Ageing produces the common feature of high heterogeneity in microglia, which is also a common phenotype of different neurodegenerative diseases [39]. Moreover, the pattern of microglial gene expression changes with ageing and neurodegenerative conditions [50]. The major phenotypic changes in ageing microglia are increased soma volume, a retraction in processes and a loss in uniform tissue distribution [51]. Furthermore, microglial activation slows with age, reducing sensing activity and impairing synaptic contact [6]. This process of ageing microglial activation is distinct from classical activation and is referred to as microglial dystrophy; the anomalous activation is more likely to be senescent rather than a classical phenotype [52,53]. Moreover, a new phenotype of microglia has been defined—dark microglia—characterised by condensed electron-dense cytoplasm and nucleoplasm, nuclear chromatin remodelling and high levels of synaptic stripping activity and oxidative stress [54,55]. Interestingly, these phenomena have not only been observed in microglial populations associated with chronic stress or diseases such as AD but also in the microglia in normally ageing brains [54]. Although knowledge of these molecular structural changes is still in its infancy, it has already been established that ageing microglia are highly granular and present an uncharacteristic dark appearance in immunohistological preparations. Another phenotypic change in aged microglia is defective lysosomal digestion. This defect largely privileges accumulation of indigestible material composed of lipofuscin and other autofluorescent pigments [56,57]. Therefore, the use of immunofluorescence or flow cytometry has become familiar among researchers to distinguish between normal and aged microglia. Accumulation of such autofluorescent pigments and lipofuscin is believed to be a by-product of impaired disposal mechanisms and purported to have a direct relation to several neurodegenerative diseases, including AD [58,59].

Microglial changes with age do not follow one specific process but, rather, change throughout one’s life; after reaching a certain age, threshold impacts will appear. A transcriptome analysis of the frontal cortex region of post-mortem healthy brains across a wide age range (from young teenagers to people over 80 years old) showed that microglial gene markers assemble into a transcriptional module in a gene co-expression network [60], and this expression pattern negatively correlates with age. Another study revealed that genes that encode microglia surface receptors for neuronal and/or microglial crosstalk are particularly affected. Several brain-expressed transcription factors, including RUNX1, IRF8, PU.1 and TAL1, are the master regulators of age-dependent microglial modulation [39]. This raises the question of how important it is to identify age-dependent genetic modulation in adulthood to understand neurodegenerative disease pathology. Identification at the beginning of genetic changes in middle or late-middle age might correlate several chronic neurodegeneration initiations, and thus may help stall disease progression.

## 4. Microglia and Neurodegenerative Diseases: Functional Relation

### 4.1. Alzheimer’s Disease

Alzheimer’s disease (AD) is the most prevalent neurodegenerative disorder, which has a characteristic feature of forming Aβ-containing plaques and neurofibrillary tangles (NFTs) containing intracellular hyperphosphorylated tau proteins and leading to neuronal loss [61] (Figure 1 and Figure 3). Microglia play a critical role in the progression and exaggeration of AD, that is, the accumulation of Aβ triggers microglia to promote tau hyperphosphorylation that eventually forms NFTs, leading to cognitive impairment. Microglia accumulate around senile plaques in AD brain parenchyma two-to-five times more than in normally functioning brains [62].

Microglia have both direct and indirect relations with AD. Evidence from genome-wide association studies (GAWS) has shown variant of *Trem2* mutations increase in risk by 3–4.5 times for developing late-onset AD [63], which is as high as the association found with *ApoE-ε4* in the AD [64]. Mutations in other microglial genes, such as *CR1*, *HLA-DRB1*, *CD33*, *MS4A6A* and *BIN1*, also have a moderate role in AD progression [64]. Although not all AD patients have similar mutant microglial gene(s), because these are core regulator genes of microglial functions, studying their roles in AD pathogenesis will affect all AD patients.

A key factor of AD pathogenesis is Aβ deposition, which is an equilibrium between Aβ production and clearance. Small changes in this production—clearance ratio result in abnormal accumulation of Aβ peptide. Therefore, microglial scavenger receptors (SRs) have an active role in Aβ clearance [65] by phagocytosis and endocytosis [66,67]. Microglia can also degrade extracellular Aβ using Aβ-degrading enzymes [34,65]. In addition, microglia from the Aβ-deposited mouse model showed reduced expression of both Aβ-phagocytic receptors and Aβ-degrading enzymes [34]. This suggests an active role of microglia in AD pathogenesis and late-onset Aβ accumulation.

However, microglia–Aβ interactions may lead to a loss of synapses [68], increased production of neurotoxic reactive oxygen and nitrogen species (ROS and RNS), activated NLRP3 inflammasomes and increased release of proinflammatory cytokines including TNF [33,69,70]. In this way, Aβ interacts with microglial pattern recognition receptors (PRRs), including Toll-like receptors (TLRs), SRs and complement receptor 3 (CR3) [24,71]. Thus, the microglial role in AD is a double-edged sword. For example, microglia sense Aβ peptides and remove these injurious agents before plaques form, whereas chronic interaction of Aβ with microglia reduces this sensing ability, resulting in further amyloid deposition. This reduction of the microglial clearing ability is a part of Aβ-induced proinflammatory cytokine production, which also activates NLRP3 and releases ASC, which binds Aβ and promotes further Aβ aggregation and spreading of amyloid pathology [70]. Therefore, while the microglia in early AD progression have a beneficial role, their malfunction will be detrimental to cells as the disease spreads. Furthermore, one study analysed the transcriptome of normal and Aβ-populated microglia and defined disease-associated microglia (DAM) [30,72]. Although the difference between DAM and dark microglia is not clear, both have a direct association with Aβ deposition and exhibit high expressions of CD11b and *Trem2* [54]. These findings indicated a direct relation between microglial transition from homeostatic to DAM during AD.

### 4.2. Parkinson’s Disease

Parkinson’s disease (PD) is the second most prevalent neurodegenerative disease, characterised by both motor and non-motor symptoms. Events of PD progression include misfolding of the protein α-synuclein and assembly into Lewy bodies and Lewy neurites, which leads to the loss of dopamine neurons in the substantia nigra region of the PD brain [73]. It has become evident from PD patients’ substantia nigra that HLA-DR-expressing reactive microglia are abundant [74]. In addition, the oligomeric α-synuclein activates microglia through heterodimer TLR1/2 and increases proinflammatory cytokine release [75]. However, the detrimental role of α-synuclein might be realised through the phagocytic receptor Axl. To support this hypothesis, a α-synuclein mutant (*SNCA^A53T^*) mice study was conducted, revealing a high level of Axl receptor expression in spinal cord analysis [26], whereas the deletion of Axl delayed disease onset. Moreover, genetic investigations of sporadic and familial PD have identified leucine-rich repeat kinase 2 (*LRRK2*) as the most common mutated gene in PD [76]. Furthermore, *LRRK2*-deficient rats showed no significant dopaminergic neuron loss and reduced myeloid cell activation in substantia nigra; this was shown in rats injected with rAAV2 α-synuclein viral particles [77].

Although the exact mechanism that could relate microglia to PD pathogenesis is not yet known, this participation is proposed to be similar to that in AD. As with Aβ clearance, microglia internalise and degrade α-synuclein to clear it. Thus, any anomalies in this process may result in an aggregation of extracellular α-synuclein [78]. These findings, although still requiring validation in animal models of PD, suggest that both AD and PD share similar pathogenic pathways. This raises the possibility that targeting microglia may also result in the double-edged sword metaphor from earlier, depending on disease progression.

### 4.3. Amyotrophic Lateral Sclerosis

Amyotrophic lateral sclerosis (ALS) is a neurodegenerative disease that causes characteristic features of progressive damage in motor neurons in the cerebral cortex, brainstem and spinal cord, potentially leading to paralysis and death. Although most patients are reported as having sporadic ALS, approximately 10% of ALS patients have mutations in specific genes, including *SOD1*, *C9orf72*, *TDP43* and *FUS74* [8]. Positron emission tomography (PET) scanning revealed increased cerebral microglia activation in ALS brains [79], and ALS brain autopsies showed that microglia are associated with the expression of proinflammatory cytokine release [80].

Furthermore, transgenic mice overexpressing G93A mutant SOD1 (mSOD1^G93A^) developed ALS-like symptoms of progressive motor neuron loss [81,82,83]. At the early onset of disease progression, microglia functions are not affected by mSOD1. Eventually, however, as the disease progresses, microglia aggravate neuronal injury by interacting with motor neurons. In ALS, microglia activate mostly through misfolding and accumulation of mSOD1. In addition, other stressors and ROS-releasing mechanisms also activate microglia and promote microglia-induced proinflammatory cytokine release. In one study, restraining microglial activation by inhibiting NF-κB signalling substantially reduced motor neuron loss and extended the survival of mSOD1^G93A^ animals [84].

Microglia change their phenotype with disease progression, accelerating disease onset [85] and exacerbating motor neuron death [86]. However, at early onset, microglia showed neuroprotective tendencies in mSOD1 mice, which increased during the late phase of disease [87]. Microglial activation and neurotoxicity in ALS are cell-autonomous processes and not only include NF-κB-dependent signalling [84], but also partly involve IL-1β [88]. Both intraneuronal and extracellular misfolded mSOD1 are sensed by microglia, and eventually microglia promote superoxide production by deregulating NADPH oxidase [89] and becoming proinflammatory [90]. Therefore, this points to the interrelation of neurodegenerative pathways between ALS, AD and PD, because in all cases where microglia sense exogenous stimuli, they respond to danger in the host and eventually change their phenotype with disease progression [8].

The expansion of hexanucleotide repeating in noncoding regions of *C9orf72* gene have displayed pathologic features of ALS in mice but have not shown behavioural abnormalities or neurodegeneration [91,92]. On the other hand, a lack of *C9orf72* causes lysosomal accumulation and increases the microglial immune response and proinflammatory activity in the host [91]. Moreover, *C9orf72* is required for maintenance of myeloid cells’ normal functioning. Although these findings seem to contradict each other, the mystery of *C9orf72* in normal microglial function has been uncovered, at least in part. Altering *C9orf72* also changes microglia-mediated misfolded protein clearance by modulating phagosome-to-lysosome maturation, suggesting that this subset may have a potential role in ALS. Therefore, future functional studies with microglia from *C9orf72* ALS patient may clarify the complexity.

Mice expressing inducible human TDP-43 (hTDP-43) showed progressive motor neuron loss, but suppressing hTDP-43 allowed microglia to clear existing hTDP-43 [93]. Interestingly, blocking microgliosis at the early recovery phase by using CSF1R and c-Kit inhibitors diminished the mice’s ability to completely retain motor functions, suggesting that microglia play a neuroprotective role [93]. In contrast, conditional deletion of TDP-43 in microglia increased their phagocytic functions and enhanced synaptic loss [94]. Thus, future studies should further investigate the link between TDP-43, microglia and ALS pathogenesis, which would help ALS patients with TDP-43 mutations by minimising dysregulation of microglial phagocytic function. This paper proposes that targeting microglia for intervening ALS should target several mutations that are associated with microglial host defence functions including *mSOD1* (ROS production), *C9orf72* (cytokines), and *C9orf72* and TDP43 (phagocytosis). Thus, targeting microglia randomly in ALS to rejuvenate associated host defence functions may not be a useful therapeutic strategy but tailoring to a specific pathway(s) could affect potentially.

### 4.4. Multiple Sclerosis

In young adults, multiple sclerosis (MS) is the most frequent neuroautoimmune disorder and is associated with severe physical nontraumatic disability. Well-defined neuroinflammatory demyelinating lesions and neuronal loss are the characteristic hallmarks of MS. Patients of this disorder begin by developing demyelinated plaques in both white and grey matter, and ongoing disease progression leads to brain atrophy and neurodegeneration. Because neuroinflammation has been evident in all stages of MS, it is proposed that the presence or absence of microglia play at least a part in inflammatory CNS of MS patients [95,96]. However, the challenge is correct distinction of resident microglia and other CNS macrophages, along with infiltrating monocyte-derived macrophages during progressive MS because all share the same surface markers and functions.

Microglia’s role in MS is unclear; they may be detrimental or beneficial. At the early onset of this disease, microglia promote axonal regeneration, clear myelin debris and release neurotrophic factors, indicating their protective role in MS [97]. However, a study of mice with experimental autoimmune encephalitis (EAE), an animal MS model, showed enhanced release of proteases, proinflammatory cytokines, ROS and RNS from microglia and recruitment of reactive T lymphocytes. Thereafter, this leads to neurotoxicity, whereas deletion of the transforming growth factor (TGF)-β-activated kinase 1 in the microglia of the EAE mouse model displayed reduced CNS inflammation. At the same time, axonal and myelin damage were reduced by the cell-autonomous inhibition of the NF-κB, JNK and ERK1/2 pathways [21], indicating that microglia may aggravate tissue injury in EAE. These results suggested that microglia actively participate at different stages of MS progression, and their role eventually changes with advancement of disease. Furthermore, it is possible that microglial function alteration is associated with specific lesions in MS, including changes in debris clearance and the neuroprotective response.

### 4.5. Huntington’s Disease

Huntington’s disease (HD) is an autosomal dominant disease, featuring progressive atrophy of the striatum and cortex [98,99]. An immunohistochemical analysis of a human HD brain showed reactive microglia present in the cortex, neostriatum and globus pallidum [100,101]. Active microglia in the striatum and cortex were also found to be related to the onset of neuronal loss [101]. In addition, a microscopic analysis of an HD model showed intranuclear inclusions containing huntingtin (HTT) protein and neurodegeneration of medium-size spiny, encephalin-containing inhibitory neurons [99]. In this condition, mutation in the HTT (mHTT) protein stretches the trinucleotide CAG and translates into polyglutamine in HTT protein sequencing, leading to HD [99].

However, microglia express HTT mRNA at a relatively high level [24], and the presence of proinflammatory microglia is correlated with a higher probability of developing HD within five years [102]. Therefore, HD severity is dependent on proinflammatory microglial onset in HD patients [101,103]. Furthermore, progressive HD changes microglia function and the genomic profile. Increased mHTT expression has been found to be linked to increases in proinflammatory genes in HD patients and in the mouse model [104], promoting myeloid linage-determining factors PU.1 and CCAAT/enhancer-binding protein (C/EBP)-α,β. This increase in transcriptional factors is correlated with the higher expression of IL-6 and TNF, and this change solely occurs in microglia [104]. Because microglia have a role as innate defensive units in their host, mHTT microglia increased several genes that sense their milieu, including *Tlr2*, *Cd14*, *Fcgr1*, *Clec4d*, *Adora3*, *Tlr9* and *Tnfrsf1b* [24,104], suggesting increase in capacity to sense extracellular stimuli. In response, the system upregulates IL-6 and TNF mRNA [104], suggesting that microglial responses have a host defence function against mHTT invasion, thereby aggravating neurodegeneration.

### 4.6. Frontotemporal Dementia

Frontotemporal dementia (FTD) is a progressive neuronal atrophy characterised by rapid neuronal loss in the frontal and temporal cortices [105]. Immunohistochemical analysis of FTD patients has shown the presence of aggregated TDP-43 in the cytoplasmic inclusions [94]. However, PET imaging of FTD patients has reported reactive microglia correlated with increased expression of proinflammatory cytokines TNF and IL-1β in the CSF [94]. Moreover, recent research documented that mutations in the *Grn* gene, encoded for the glycoprotein progranulin, lead to FTD. In addition, progranulin is mainly expressed by neurons and microglia in the CNS [106]. In pathological conditions, microglia upregulate progranulin; however, it is assumed that progranulin deficiency might impair autophagy and lead to FTD progression [106]. In support of this, progranulin deletion in mice showed increased microgliosis [107]. Based on these results, this study proposed that microglia have a potential role in FTD, and, as in other neurodegenerative diseases, they should be evaluated by genomic study in the future.

### 4.7. Prion Disease

Prion diseases include spongiform encephalopathies and are due to the gradual aggregation of the prion protein PrP^Sc^ [7]. In this vein, PrP^Sc^ can transmit itself into new hosts as an exogenous seed that can cause protein misfolding and aggregation, as well as aggravate the disease in the absence of microbial agents, viruses or inflammation. This feature is referred to as the prion paradigm and has been evident in cases of other neurodegenerative diseases, including AD, PD, ALS and FTD [7,108]. Prion-related neurodegeneration includes increased neuronal loss and proinflammatory microglia. Microglia and other CNS macrophages phagocytose PrP^Sc^ approximately 60 days earlier after sensing infection [109], but their depletion increases prion infection susceptibility [110], suggesting that microglia play a major role in controlling prion disease. The double-edged sword metaphor applies here as well for microglia because they produce ROS in response to the PrP106-126 fragment and promote PrP-induced neurotoxicity. Suppressing the superoxide-producing enzymes produced protective results from PrP-induced toxicity in mice, further suggesting that microglia mediate prion neurodegeneration [111].

Although it is unclear whether or not microglial proinflammatory releases affect prion disease progression, several proinflammatory cytokines—IL-1β, IL-6, inducible nitric oxide synthase (iNOS), NF-κB, cyclophilin A, matrix metalloproteinases and NLRP3 inflammasome components—have been upregulated in prion disease microglia [112,113]. It has also been shown that prion infection affects microglial sensing and housekeeping ability through the disruption of the Cx3cr1–fractalkine pathway [114]. Prion protein PrP^Sc^ impairs microglial ability to phagocytose aberrant proteins, including PrP^Sc^ and apoptotic debris or cells. In addition, this impairment promotes the microglial proinflammatory mediator’s production, which dysregulates host defence [115]. Microglial function regarding PrP^Sc^ is SR- and TLR-mediated in an Src-kinase-dependent manner [116,117], which suggests that microglia might initially engage in PrP^Sc^ clearance, but their consistent malfunctions and activation in another example impair the host-defensive response. Thus, normal microglial functions result in neurotoxic action and subsequently aid in disease progression.

Thus far, two common themes for microglia involvement in different neurodegenerative diseases have emerged. First, microglia perform their regular sentinel function after sensing the aberrant or misfolded proteins such as Aβ, aggregated α-synuclein, oxidised or mutant SOD1, or PrP^Sc^ (Figure 1). Next, they attempt to clear those toxic stimuli via SRs and/or other PRRs as their host-defence function. However, persistent production of aberrant proteins reduces microglial host-defence regulatory functions and dysregulates microglial immune checkpoints that keep microglia-induced inflammation in control, such as the Cx3cr1 or progranulin pathways. Thus, normal microglia are led into a proinflammatory state and a further response in the host defence through exaggerated neuroinflammation and neurodegeneration. Second, some neurodegenerative diseases cause self-autonomous actions; mutations in specific genes, such as *Trem2*, HTT and TDP43, and dysregulate host abilities of sensing, housekeeping and defence. In this way, microglial actions initiate or exaggerate neurotoxicity and neurodegeneration.

## 5. Microglia as a Therapeutic Possibility in Neurodegenerative Diseases

Microglia play vital roles at different brain development phases. With age or due to aberrant endogenous or exogenous stimuli, they begin losing their normal physiological functions. Thus, homeostasis in CNS microglia is necessary to disrupt neurodegenerative disease pathology and progression. In addition, identification of critical microglial markers is important to find new therapeutic strategies. Initial studies have suggested an M1 and M2 activation paradigm, in which M1 activation promotes inflammatory cytokines, and M2 activation promotes neurotrophic factor release. However, advanced studies have found that this paradigm does not fit during neurodegeneration. That is, M1/M2 activation does not always function as expected. For instance, in regular cases, M1 activation produces neurotoxicity via proinflammatory release, whereas in some cases, this activation promotes axonal regeneration [118]. An AD model study showed that M1 activation promotes Aβ plaque clearance, and, in contrast, M2 activation may ease amyloid spread [119].

Microglia are emerging as a cell type used to understand neurodegenerative diseases, but the major challenge is studying human microglia in vitro. For an in vitro neurological disease model study, microglial cells could possibly be developed by differentiating iPSCs or monocytes [120,121], which has recently been demonstrated. In addition, authors have shown that human microglial-like cells (iMGLs) have phenotypic similarity to in vivo microglia, such as through inflammatory cytokine release or CNS substrate phagocytosis. On the other hand, human monocyte-derived, microglia-like (MDMi) cells have not only presented with microglia phenotype and functions but have also presented with altered expression of gene loci related to neurodegenerative diseases such as AD, PD and MS. These two in vitro microglial cell models could beneficiate therapeutics screening in vitro for neurodegenerative diseases. In addition, genetic defects in microglia could be edited by replacing allogenic or autologous stem cells or monocytes through bone marrow transplantation. Although the latter has not been successfully established for all neurological diseases—though it has for X-linked adrenoleukodystrophy—a recent study showed that brain-engrafted bone marrow derived microglia after a long time in AD mice [122,123]. It has been suggested that peripheral myeloid cells constitute a heterogeneous cell population that is more effective at clearing Aβ plaque than CNS resident microglia. Extrapolating this therapy with additional triggering could bring success or could be useful for studying other neurological diseases.

Microglial phagocytosis could be another option, but therapeutic agents that target microglial phagocytosis can have both beneficial and detrimental effects—another double-edged sword in neurodegeneration. Microglial phagocytosis opsonises misfolded protein plaques, including Aβ, via the Fc receptor to help antibodies that target misfolded proteins [7,124,125]. In the same way, defective microglia activation in *Trem2*-deficient mice showed a lack of effectiveness toward the anti-Aβ antibodies [126]. Several antibodies currently used in autoimmune diseases may be beneficial in neurodegenerative diseases as well because they target specific proinflammatory cytokines, such as IL-6 and IL-1, or their receptors. Compounds targeting CSF1R can affect proinflammatory microglia activation in AD [127,128] and reduce microglia-induced inflammation and/or neuronal death [129] in neurodegenerative diseases. In the same vein, IL-34 and CSF1, ligands of CSF1R, may provide neuroprotection and promote neuronal cell survival shown in neurodegenerative models by activating CSF1R in neuron populations but not in microglia [130].

In addition, bexarotene-induced *Trem2* expression in microglia is, at least in part, mediated by *ApoE*/*Trem2* signalling activation [131]. Thus, developing anti-*ApoE* antibodies in carriers of the ApoE4 allele may help to prevent amyloid deposition and its consequences [132]. Moreover, neuronal autophagy has been shown to be useful in neurodegenerative diseases for clearing misfolded proteins and reducing inflammatory cytokines [133]. However, very little is known about the role of microglia in autophagy. In one study, loss-of-function mutations of TBK1 affected autophagy in myeloid cells and increased susceptibility to ALS [134]. Beyond this, microglial autophagy facilitates Aβ clearance and reduces NLRP3 inflammasome activation [135]. Therefore, further study of autophagy in microglia may enhance the understanding of whether drugs activating autophagy have beneficial or detrimental impacts on neurodegenerative diseases.

## 6. Conclusions

Microglial biology has gained substantial attention in recent decades. Several advancements have been introduced, including microglial gene expression checks, longevity analysis in a single cell in the neurodegenerative disease model, pathways that regulate their responses to neuronal injury, pathways that check microglial inflammatory responses and pathways that promote injurious stimuli clearance. In addition, advanced research has shown how peripheral influences from the gut microbiome can alter such injuries. However, substantial knowledge gaps exist that slow the therapy-finding process through the microglial regulated pathway. One major limitation in this process is a reliable disease model. More reliable cellular in vitro disease models and the addition of new technologies for in vivo modelling for better imaging and analysis could strengthen understanding of microglial involvement in neurodegeneration. Furthermore, analysis of the transcriptomes and epigenetic profiles in various diseases shows that it is essential to understand the relevance of ageing and disease progression in relation to the alteration in these profiles at the single-cell level and to thereafter correlate such changes with microglial behaviour. Finally, these steps could bring about a crucial breakthrough in microglia-mediated therapeutic intervention in neurodegenerative diseases.

## Figures and Tables

**Figure 1 cells-10-00150-f001:**
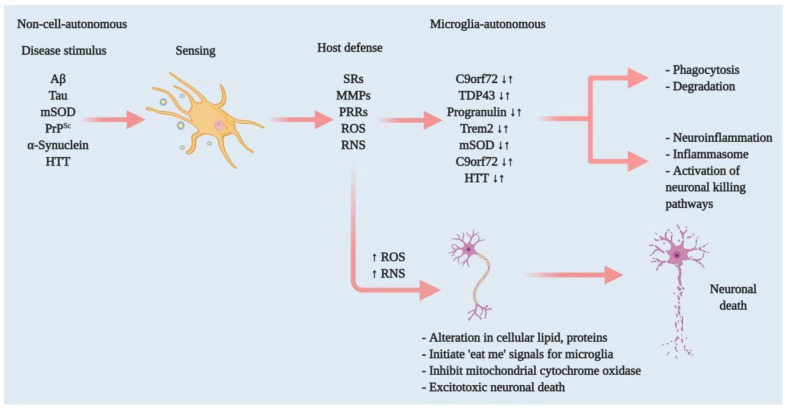
Effectors of microglia function associated with neurodegeneration and how microglia damage or kills neurons. Microglia have both host-defensive and detrimental actions, depending on the scenario. When they encounter aberrant or misfolded proteins—such as Aβ, aggregated α-synuclein, oxidised or mSOD1, or PrP^Sc^—they function as sentinels, protective hosts. In response to these toxic stimuli, microglia attempt to clear aggregated Aβ, α-synuclein or mSOD1 via SRs and other PRRs. However, the nature of the aberrant proteins or their continued production disrupts microglial housekeeping functions. Several pathways activate: NADPH oxidase produces superoxide and derivative oxidants; iNOS produces nitric oxide derivatives, glutamate, cathepsin B and other proteases, and that produces stressed neurons. These processes dysregulate the microglial host-defensive mechanism, leading to an exaggerated proinflammatory response, neurotoxicity and neurodegeneration. Microglia also include tumour necrosis factor (TNF) as an indirect pathway to damage or kill neurons via reducing brain-derived neurotrophic factor (BDNF) and insulin-like growth factor (IGF) production. Neurodegenerative diseases such as AD, ALS and HD, furthermore, cause mutations in specific genes that lead to self-autonomous dysregulation of host defence. This initiates or exaggerates proinflammatory responses, resulting in neurotoxicity and neurodegeneration. In this way, when mutations in TDP-43, progranulin and *Trem2* increase (↑), they affect phagocytosis and associated degradation pathways. Similarly, mutations in mSOD and HTT (↑) also affect inflammasome activation and neuronal killing pathways. On the other hand, mutations in *C9orf72* can affect both phagocytosis and inflammasome pathways. In normal physiology, a microglia-autonomous mechanism controls the scenario by clearing (↓) mutant or aberrant proteins (adapted from ref. [8]).

**Figure 2 cells-10-00150-f002:**
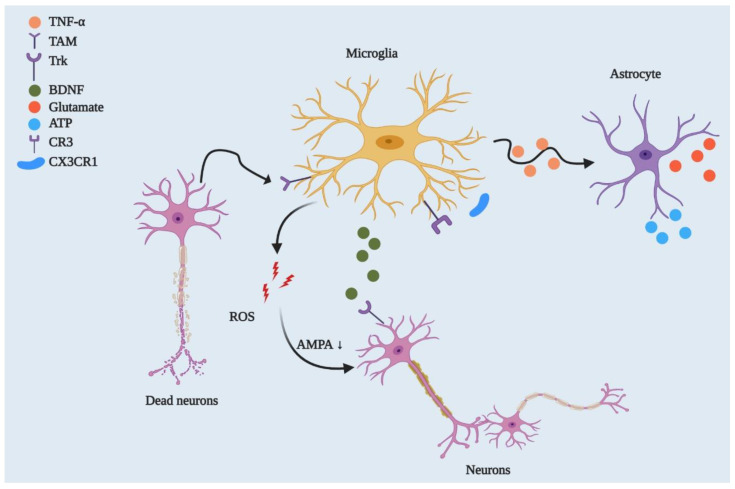
Microglial regulation of neuronal networks and CNS homeostasis. By playing a phagocytic role, microglia engulf dead or degenerated neurons via TAM receptor-mediated recognition of GAS6-opsonised cells. Microglia control synaptic plasticity by downregulating AMPA receptor activity through ROS secretion and by modulating BDNF/Trk receptor pathways. Beyond this, microglia can strip synapses through *Cx3cr1-Cx3cl1* interactions and microglial major histocompatibility complex II (MHCII) [32]. They can also affect neuronal circuitry via interacting with astrocytes through the TNF-mediated pathway, which releases glutamate and ATP from astrocytes.

**Figure 3 cells-10-00150-f003:**
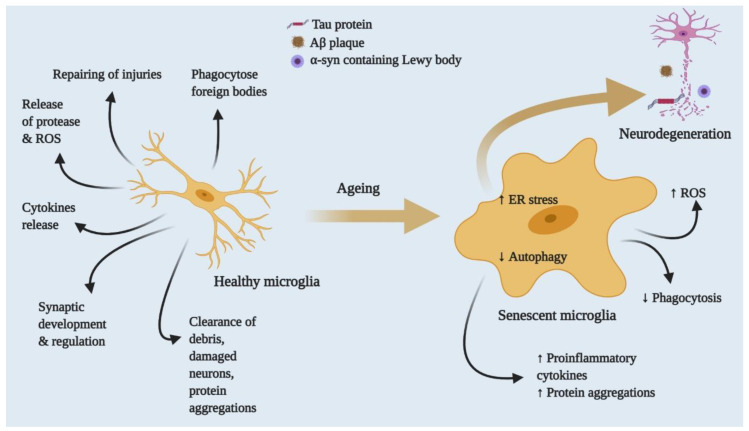
Effects of ageing on healthy to proinflammatory microglia. Healthy microglia regulate the release of cytokines, proteases, ROS and RNS and phagocyte foreign bodies. In addition, they clear cell debris and proteins. With advanced ageing, microglia eventually lose function or functions dysregulated by persistent exposure to foreign bodies or mutations. Increasing microglial ROS production releases proteases and cytokines and slows the autophagic process, leading to aberrant protein aggregation and inflammatory response, neuronal death and neurodegenerative disease.

## Data Availability

Not applicable.

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
