# Peer review of "Microglial Turnover in Ageing-Related Neurodegeneration: Therapeutic Avenue to Intervene in Disease Progression"

_cells, 2021, doi:10.3390/cells10010150_

Round 1

Reviewer 1 Report

The review by Azam et al focuses on the role of microglia in neurodegenerative diseases associated with ageing. Although the issue is not so novel, the paper is valuable and well written and the figures very nice and complete.

I suggest expanding the paragraph about drugs acting on microglia-related mechanisms as potential therapeutics in neurodegeneration, also adding more recent papers, and discuss possible applications in clinical trials.

Author Response

Response to reviewer comments

Title:            Microglial turnover in ageing-related neurodegeneration: Therapeutic avenue to intervein in disease progression

Journal:       Cells

Ms. ID:        cells-1030993

Reviewer 1

The review by Azam et al focuses on the role of microglia in neurodegenerative diseases associated with ageing. Although the issue is not so novel, the paper is valuable and well written and the figures very nice and complete.

I suggest expanding the paragraph about drugs acting on microglia-related mechanisms as potential therapeutics in neurodegeneration, also adding more recent papers, and discuss possible applications in clinical trials.

Response: We appreciate this suggestion; therapeutic avenues are one prime focus of this manuscript. We have tried to link several possible therapeutic options from current knowledge. Following reviewer’s suggestion, we have added additional paragraph briefing possible solution of genetic defects in microglia in neurodegenerative diseases. Besides, we have suggested other therapeutic options throughout the article, where applicable. We thank again for this valuable comment.

We appreciate the time given by the reviewers towards our manuscript, and their interests in our article. All the comments raised are of great importance for making this article more informative and supportive for readers in this research area.

Reviewer 2 Report

Review: This review is focusing on possible contributions of microglia to several neurodegenerative disease. It seems to be a rather comprehensive review. However, due to language issues it is very difficult to evaluate science described in this review. Below are just a few examples where statements were inaccurate or their meaning was not clear. Recommendation is that manuscripts is first thoroughly edited for English so that review on the science content can be possible.

Several statements are not accurate -these are just few examples not complete list:

Under healthy and normal conditions microglia are maintained by the peripheral myeloid cells”-This is not accurate brain microglia are self-maintained under normal conditions.

“During development phase of brain, microglia help shaping neural circuits by modulating strength of synaptic transmissions and forming neuronal synapse”- During development microglia were shown to play a key role in eliminating synapses in activity dependent manner.

In response to CNS insult, microglia proliferate rapidly [9], their rapid proliferation and turnover under homeostatic conditions could be a point to define their characteristics properly [10}” While microglia proliferate in some types of CNS insult , not all CNS insults induce proliferation-depending on the strength of the insult, some responses may include only morphological and gene expression alterations in microglia.

Example of statements whose meaning is not clear:

“Microglia are one major brain-residing macrophage [1] that is distinct from other residing macrophagic populations [2,3]”. –Are authors here referring to other tissues macrophages or other brain macrophages?

Development of microglial cells could be possible by reprograming stem cells or monocytes [24,25], which has been demonstrated recently. –Reviewer assumes that here authors refer to creating/differentiating microglia in vitro from iPSCs? It is confusing as this is in the paragraph about microglial development.

“For that, microglia develops a thin network that bridge throughout the CNS [19]. This network includes around 100 genes or more that consistently scans surrounding cell bodies and senses any changes in their microenvironment [26-28]. The microenvironment is called sensome and it has been seen that mRNAs of sensome are widely distr…”

It is unclear what authors mean by thin network? And sensome is a term referring to cluster of genes encoding proteins for sensing endogenous ligands and microbes and not microenvironment.

Author Response

Response to reviewer comments

Title:            Microglial turnover in ageing-related neurodegeneration: Therapeutic avenue to intervein in disease progression

Journal:       Cells

Ms. ID:        cells-1030993

Reviewer 2

Review: This review is focusing on possible contributions of microglia to several neurodegenerative disease. It seems to be a rather comprehensive review. However, due to language issues it is very difficult to evaluate science described in this review. Below are just a few examples where statements were inaccurate or their meaning was not clear. Recommendation is that manuscripts is first thoroughly edited for English so that review on the science content can be possible.

Response: Thank you for this recommendation, we have edited this manuscript by native speaker.

Several statements are not accurate -these are just few examples not complete list:

Under healthy and normal conditions microglia are maintained by the peripheral myeloid cells”-This is not accurate brain microglia are self-maintained under normal conditions.

Response: Thank you for this comment, microglia are brain macrophage, and they are different from other tissue-resident macrophage populations. Their maintenance is self-regulated process with no considerable contribution of peripheral myeloid cells, under normal conditions. We have corrected the sentence in revised version.

“During development phase of brain, microglia help shaping neural circuits by modulating strength of synaptic transmissions and forming neuronal synapse”- During development microglia were shown to play a key role in eliminating synapses in activity dependent manner.

Response: We appreciate this comment, microglia plays a vital role in neuronal circuitry shaping during development phase. During that time, they sculpt synapses not forming, which we tried to mean in that sentence but typed wrong word. We have corrected in revised version.

In response to CNS insult, microglia proliferate rapidly [9], their rapid proliferation and turnover under homeostatic conditions could be a point to define their characteristics properly [10}” While microglia proliferate in some types of CNS insult, not all CNS insults induce proliferation-depending on the strength of the insult, some responses may include only morphological and gene expression alterations in microglia.

Response: We appreciate this concern and agree that not all stimuli can induce rapid proliferation of microglia due to strength. This sentence refers to neurodegenerative disease state when they become reactive because of chronic exposure to different stimuli.

Example of statements whose meaning is not clear:

“Microglia are one major brain-residing macrophage [1] that is distinct from other residing macrophagic populations [2,3]”. –Are authors here referring to other tissues macrophages or other brain macrophages?

Response: Thank you for this comment, here we meant other “brain” -residing macrophages. Microglia are the only macrophage population in the central nervous system (CNS) parenchyma, where there are other non-parenchymal macrophages are also present in the brain, including meningeal, perivascular and choroid plexus macrophages. Although non-parenchymal macrophages are less well defined, but they express different genes and play important functions in the blood-brain barrier and blood-cerebrospinal fluid barrier. They are distinct from parenchymal macrophage, microglia. On the other hand, tissue macrophages are functionally distinct from brain macrophages. We have corrected the sentence in revised manuscript.

Development of microglial cells could be possible by reprograming stem cells or monocytes [24,25], which has been demonstrated recently. –Reviewer assumes that here authors refer to creating/differentiating microglia in vitro from iPSCs? It is confusing as this is in the paragraph about microglial development.

Response: We appreciate this comment and apologies for misplacing the information. Although microglial functions study is a key to understand different neurological diseases, but still in vitro human microglial studies are challenging. As our manuscript focus was to analysis current understanding of microglial functions and challenges to link them in different neurological diseases, we tried to introduce newer in vitro models so that reader may benefit from it. However, both of our cited articles have studied human microglial-like cell models, which they have differentiated from either iPSCs or monocytes and used them into in vitro and in vivo, and their findings showed phenotypic similarity to microglia. We have replaced this information in the therapeutic paragraph in our revised version.

“For that, microglia develops a thin network that bridge throughout the CNS [19]. This network includes around 100 genes or more that consistently scans surrounding cell bodies and senses any changes in their microenvironment [26-28]. The microenvironment is called sensome and it has been seen that mRNAs of sensome are widely distr…”

It is unclear what authors mean by thin network? And sensome is a term referring to cluster of genes encoding proteins for sensing endogenous ligands and microbes and not microenvironment.

Response: Thank you for this comment, microglia forms a network to convey their response or scan surrounding microenvironment constantly and rapidly. This process is weak and dynamic and constantly in motion, so we mean the process through the word “thin network”, we have modified the sentence in revised version to mean clearly what we intend to mean.

Sensome usually refers to the group of unique protein transcript used to sense ligands and microbes. But it is also referring to the genes required for the proteins used to sense molecules within the body. Reviewer is right about the sentence, we apologies for that mistake and corrected the sentence in revised version.

We appreciate the time given by the reviewers towards our manuscript, and their interests in our article. All the comments raised are of great importance for making this article more informative and supportive for readers in this research area.

Round 2

Reviewer 2 Report

Authors have significantly improved the manuscript. However there are still several statements that need to be clarified/corrected to improve the readability and accuracy of the manuscript. Here are some examples:

  1. Neurodegenerative diseases such as AD, ALS and HD, furthermore, cause mutations in specific genes that lead to self-autonomous dysregulation of host defence.

Can authors please clarify this sentence: i.e. do they mean that disease pathogenesis of AD, ALS and HD includes causing mutations in specific genes?

  1. Their thin processes are dynamic and in constant motion, allowing them to scan the area surrounding their cell  body every few hours and rapidly polarize toward focal injury [19].

Can authors please clarify what do they mean by “allowing them to scan the area surrounding their cell body every few hours” and cite the manuscript showing that?

  1. eyond this, microglia can strip synapses through Cx3cr1-Cx3cl1 interactions”

For synaptic removal by microglia complement system and MHCII have been very well studied but have not been mentioned by authors here.

  1. Identification at the beginning of genetic changes in middle or late-middle age might correlate several chronic neurodegeneration initiations and,thus, may help stall disease progressi

Please clarify.

  1. Evidence from genome-wide  association studies (GAWS) has shownTrem2 mutations increase in riskby 3–4.5 times fordeveloping AD [62], which is as high as their association found with ApoE-ε4 [63].

GWAS? The association?

  1. and increased release of proinflammatory cytokines and TNF [32, 68,

TNF alpha is a proinflamatory cytokine,

  1. For example, microglia sense Aβ peptides and remove these injurious agents before plaques form, whereas chronic interaction of Aβ with microglia reduces this sensing ability, resulting in further amyloid deposition. This reduction of the microglial clearing ability is a part of Aβ-induced proinflammatory cytokine production, which also activates NLRP3 and releases ASC, which binds Aβ and promotes further Aβ aggregation and spreading of amyloid pathology [69].

Can authors clarify whether by “sense Aβ peptides and remove these injurious agents before plaques” they mean microglia express receptors that bind and phagocytose Aβ peptides? If this is the case –do they then propose that “chronic interaction of Aβ with microglia reduces this sensing ability” reduces expression of that receptor?

  1. Therefore, until the microglia in early AD progression have a beneficial role, their malfunction will be detrimental to cells as the disease spreads”

Do authors want to use-while instead highlighted until?

  1. Furthermore, one study analysed the transcriptome of normal and Aβ populated microglia and defined disease-associated microglia (DAM) [30, 71]

Can authors please clarify “Aβ populated microglia”?

  1. Microglia activate several receptors to incite host defences, such as expressing Fc receptors, Toll-like receptors (TLRs), and several viral or antimicrobial peptides [24].

Can authors clarify whether they are stating that microglia activate “several viral or antimicrobial peptides”?

  1. Moreover, genetic investigations of α-synuclein have identified leucine-rich repeat kinase 2 (LRRK2) as the most common mutated gene in PD [75]

Can authors please clarify what they mean by “genetic investigations of α-synuclein”?

  1. Furthermore, LRRK2-deficient rats showed no significant dopaminergic neuron loss or reduced myeloid cell activation in substantia nigra [76]

Perhaps authors want to use and instead of or here? Also it is important to add that this was shown in rats injected with rAAV2 α-synuclein viral particles.

  1. Positron emission tomography (PET) scanning revealed cumulates of microglia surrounding injured neurons in ALS brains [78],

Reviewer does not think that resolution of PET allows for detecting cell types interactions such as above described microglia surrounding injured neurons. Could authors please recheck this in ref 78?

  1. Protein mutations in microglia through the expansion of hexanucleotide repeatinginnoncoding regions of C9orf72 gene have displayed pathologic features of ALS in mice but have not shown behavioural abnormalities or neurodegeneration [90, 91].

Could authors please clarify this statement i.e. which protein mutations are they discussing, are the mice expressing expanded C90rf72 only in microglia?

  1. This paper proposes that targeting microglia for intervening ALS may be a specific mutant gene oriented to rejuvenate microglial host defence functions, including the production of ROS (mSOD1), cytokines (C9orf72) and phagocytosis 319 (C9orf72 and TDP43).

Suggest rewriting.

  1. In addition, a microscopic analysis of an HD model showed intranuclear inclusion of Huntington (HTT) protein and enkephalin containing inhibitory neurons [98].

Can authors please clarify are they stating that intranuclear inclusion of Huntington (HTT) are in enkephalin containing inhibitory neurons?

  1. Because microglia have a role as innate defensive units in their host, mHTT microglia sense several extracellular stimuli, including Tlr2, Cd14, Fcgr1, Clec4d, Adora3, Tlr9 and Tnfrsf1b [24, 103].

Can authors please clarify are they stating that “microglia sense extracellular stimuli including Tlr2, Cd14, Fcgr1, Clec4d, Adora3, Tlr9 and Tnfrsf1b”, meaning that Tlr2, Cd14, Fcgr1, Clec4d, Adora3, Tlr9 and Tnfrsf1b are extracellular stimuli?

  1. In response, the system upregulates IL-6 and TNF mRNA [103], suggesting that microglial responses have a host defence function against mHTT invasion, thereby aggravating neurodegeneration.

Could authors please clarify?

  1. Microglia and other CNS macrophages phagocytose PrPSc approximately60 days earlier after sensing infection [108], but their depletion increases prion infection susceptibility

Could authors please clarify “60 days earlier after”?

  1. Could authors please clarify these statements:

 Second, some neurodegenerative diseases are caused by cell-autonomous action; mutations in specific genes, such as Trem2, HTT and TDP43; and dysregulating host abilities of sensing, housekeeping and defence.

Thus, homeostasis in CNS microglia is necessary to disrupt neurodegenerative disease pathology and progression.

Author Response

Authors have significantly improved the manuscript. However there are still several statements that need to be clarified/corrected to improve the readability and accuracy of the manuscript. Here are some examples:

  1. Neurodegenerative diseases such as AD, ALS and HD, furthermore, cause mutations in specific genes that lead to self-autonomous dysregulation of host defence. Can authors please clarify this sentence: i.e. do they mean that disease pathogenesis of AD, ALS and HD includes causing mutations in specific genes?

Response: Thank you for this concern, with this sentence we meant AD, ALS and HD pathogenesis includes specific genes mutation. Microglia has two distinct roles in neurodegenerative diseases, first their normal sentinel function impaired due to chronic exposure to misfolded or aberrant protein and second mutations in specific genes cause hosts defence impairment self-autonomously. For example, mutations in TDP-43 or Trem2 affect microglial phagocytosis and related degradation or mutation in mSOD and HTT affect inflammasome activation and neuronal killing pathways.

  1. Their thin processes are dynamic and in constant motion, allowing them to scan the area surrounding their cell body every few hours and rapidly polarize toward focal injury [19]. Can authors please clarify what do they mean by “allowing them to scan the area surrounding their cell body every few hours” and cite the manuscript showing that?

Response: Appreciate this concern, we have added citation where there is a histogram of normal mouse brain showing microglia moving constantly and spanning network. They have stained with anti-CD11b as microglia marker and provided a video as supplementary to show how microglia spanning network and scanning hourly.

  1. eyond this, microglia can strip synapses through Cx3cr1-Cx3cl1 interactions”. For synaptic removal by microglia complement system and MHCII have been very well studied but have not been mentioned by authors here.

Response: Appreciate this concern, major histocompatibility complex II (MHCII) is a macrophagic marker expressed by microglia when activated or in pathological context. Microglial MHCII also have significance in normal physiological functions. Indeed, inclusion of the physiological role of microglial MHCII is important for this article, but unfortunately, we missed. Appreciating this comment and considering as kind suggestion for our future study designing on microglia. In current manuscript, we mentioned once with citation.

  1. Identification at the beginning of genetic changes in middle or late-middle age might correlate several chronic neurodegeneration initiations and, thus, may help stall disease progressi Please clarify.

Response: Thank you for this comment, genetic mutations are common in some neurological diseases such as AD, PD, HD or ALS. These mutations also affect microglial normal physiological functions. Although, several recent research have specified several mutant protein in the context of disease pathogenesis, but still no clear clue that at what part of ageing these mutations start. Through this sentence we want to mean if we can identify the triggering point or age range when the mutations start, we may intervein in disease progression or initiations.

  1. Evidence from genome-wide association studies (GAWS) has shownTrem2 mutations increase in riskby 3–4.5 times fordeveloping AD [62], which is as high as their association found with ApoE-ε4 [63]. GWAS? The association?

Response: Appreciate the concern, GWAS is an approach that is usually used in genetic research to find association of a specific gene with a particular disease. The ApoE gene is a major risk factor for AD. Our cited article has conducted a 2 stage meta-analysis in native European populations and identified 11 new loci associated with AD. In that article they referred to a recent article that showed a missense mutation in TREM2 is associated with late-onset AD. Besides, TREM2-ApoE pathway is a major regulator of microglial functional phenotype in neurodegenerative diseases. This sentence tried to mean variant in TREM2 carries as much risk of developing AD as carried by the ε4 allele of ApoE gene in AD. We have corrected the sentence.

  1. And increased release of proinflammatory cytokines and TNF [32, 68, TNF alpha is a proinflamatory cytokine,

Response: Thank you for this comment, we have corrected the sentence.

  1. For example, microglia sense Aβ peptides and remove these injurious agents before plaques form, whereas chronic interaction of Aβ with microglia reduces this sensing ability, resulting in further amyloid deposition. This reduction of the microglial clearing ability is a part of Aβ-induced proinflammatory cytokine production, which also activates NLRP3 and releases ASC, which binds Aβ and promotes further Aβ aggregation and spreading of amyloid pathology [69]. Can authors clarify whether by “sense Aβ peptides and remove these injurious agents before plaques” they mean microglia express receptors that bind and phagocytose Aβ peptides? If this is the case –do they then propose that “chronic interaction of Aβ with microglia reduces this sensing ability” reduces expression of that receptor?

Response: Thank you for this comment, Aβ peptides are detected by the microglial pattern-recognition receptors such as Toll-like receptors (TLRs) and might be evolved as a host response. While many of the inflammatory mediators released upon the microglial activation and might aid in the function of the brain and initially support the clearance of pathogenic Aβ. In contrast to an immune response to initial insults, which is terminated once the stimulating pathogen has been removed, sustained elevation of Aβ and continuous Aβ aggregation does not allow the resolution of inflammation but instead fuels a chronic reaction of the innate immune system. In this sentence, using the word “sense” we meant microglial normal physiological functions, which is impaired upon consistent exposure to pathogenic Aβ or aggregated Aβ.

  1. “Therefore, until the microglia in early AD progression have a beneficial role, their malfunction will be detrimental to cells as the disease spreads” Do authors want to use-while instead highlighted until?

Response: Thank you for this suggestion. We have corrected the sentence.

  1. Furthermore, one study analysed the transcriptome of normal and Aβ populated microglia and defined disease-associated microglia (DAM) [30, 71]. Can authors please clarify “Aβ populated microglia”?

Response: Appreciate this concern, authors of the cited article have identified a special type of microglia that is associated with neurodegeneration and possibly protecting brain from progressive Aβ pathogenesis. In that process they stained DAM to identify their location and found them close to Aβ plaques or where intracellular Aβ particles are dense, so using the term “Aβ populated microglia” we meant DAM location.

  1. Microglia activate several receptors to incite host defences, such as expressing Fc receptors, Toll-like receptors (TLRs), and several viral or antimicrobial peptides [24]. Can authors clarify whether they are stating that microglia activate “several viral or antimicrobial peptides”?

Response: Thank you for this concern, the cited article has found several unique transcripts are expressed by microglia including antimicrobial peptides Camp and Ngp. We apologise for the use of the term viral that have been deleted in revised version.

  1. Moreover, genetic investigations of α-synuclein have identified leucine-rich repeat kinase 2 (LRRK2) as the most common mutated gene in PD [75]. Can authors please clarify what they mean by “genetic investigations of α-synuclein”?

Response: Apology for misuse of the term “α-synuclein”, it should be sporadic and familial PD, which has been corrected.

  1. Furthermore, LRRK2-deficient rats showed no significant dopaminergic neuron loss or reduced myeloid cell activation in substantia nigra [76]. Perhaps authors want to use and instead of or here? Also it is important to add that this was shown in rats injected with rAAV2 α-synuclein viral particles.

Response: Appreciate this concern, we have corrected the sentence according to the suggestion.

  1. Positron emission tomography (PET) scanning revealed cumulates of microglia surrounding injured neurons in ALS brains [78], Reviewer does not think that resolution of PET allows for detecting cell types interactions such as above described microglia surrounding injured neurons. Could authors please recheck this in ref 78?

Response: Appreciate this concern, the cited article has measured binding potential of different brain regions including motor cortex, pons and dorsolateral prefrontal cortex targeting a ligand PK11195 for the peripheral benzodiazepine site expressed by activated microglia. From their findings they suggested that cerebral microglial activation could be detected in vivo during ALS progression and their findings is well supported by previous findings on cerebral pathology. We have modified that sentence, to clearly express the information.

  1. Protein mutations in microglia through the expansion of hexanucleotide repeatinginnoncoding regions of C9orf72 gene have displayed pathologic features of ALS in mice but have not shown behavioural abnormalities or neurodegeneration [90, 91]. Could authors please clarify this statement i.e. which protein mutations are they discussing, are the mice expressing expanded C90rf72 only in microglia?

Response: Thank you for pointing this sentence, here we tried to interpret an article (ref. 91) that showed how important C9orf72 protein is in endosomal trafficking in macrophages. Their mouse model with expanded C9orf72 expressed in macrophages, dendritic cells and some other immune cells, but highest expression was in macrophages. From their findings we understand that C9orf72 is necessary for normal functioning of myeloid cells and microglia, and C9orf72 expanded carriers may alter microglial function and contribute to neurodegeneration including ALS.

No, expanded C9orf72 did not expressed only in microglia. We have modified the sentence.

  1. This paper proposes that targeting microglia for intervening ALS may be a specific mutant gene oriented to rejuvenate microglial host defence functions, including the production of ROS (mSOD1), cytokines (C9orf72) and phagocytosis 319 (C9orf72 and TDP43). Suggest rewriting.

Response: Thank you for this suggestion, we have modified the sentence.

  1. In addition, a microscopic analysis of an HD model showed intranuclear inclusion of Huntington (HTT) protein and enkephalin containing inhibitory neurons [98]. Can authors please clarify are they stating that intranuclear inclusion of Huntington (HTT) are in enkephalin containing inhibitory neurons?

Response: Appreciated this concern, in this sentence we have tried to state that HD patient have intracellular inclusion and that inclusion contains HTT protein. Second part of this sentence meaning neurodegeneration in HD patients are associated with medium-size spiny, enkephalin containing inhibitory neurons. We have modified the sentence, so that it express exact information.

  1. Because microglia have a role as innate defensive units in their host, mHTT microglia sense several extracellular stimuli, including Tlr2, Cd14, Fcgr1, Clec4d, Adora3, Tlr9 and Tnfrsf1b [24, 103]. Can authors please clarify are they stating that “microglia sense extracellular stimuli including Tlr2, Cd14, Fcgr1, Clec4d, Adora3, Tlr9 and Tnfrsf1b”, meaning that Tlr2, Cd14, Fcgr1, Clec4d, Adora3, Tlr9 and Tnfrsf1b are extracellular stimuli?

Response: Thank you for pointing this sentence, we have modified to make clear what we intended to mean.

  1. In response, the system upregulates IL-6 and TNF mRNA [103], suggesting that microglial responses have a host defence function against mHTT invasion, thereby aggravating neurodegeneration. Could authors please clarify?

Response: Thank you, here we tried to mean that expression of mHTT in microglia increases proinflammatory genes, which is a cell-autonomous process. As a part of microglial defence mechanism, they increase several genes expression after sensing any extra or intracellular stimuli that are threat to cells. In that process they also increase proinflammatory cytokines release including IL6 and TNF. Thereby the microglial neurotoxicity observed in association with HD are actually microglial exaggerated host-defence response that tries to get rid of mHTT but ends with worsen neurodegeneration.

  1. Microglia and other CNS macrophages phagocytose PrPSc approximately60 days earlier after sensing infection [108], but their depletion increases prion infection susceptibility Could authors please clarify “60 days earlier after”?

Response: Thank you, here we tried to mean that microglia phagocytose PrPSc as early as 60 days postinfection.

  1. Could authors please clarify these statements:

Second, some neurodegenerative diseases are caused by cell-autonomous action; mutations in specific genes, such as Trem2, HTT and TDP43; and dysregulating host abilities of sensing, housekeeping and defence.

Response: Some neurodegenerative diseases includes mutations in specific genes including Trem2, HTT and TDP3, and causes a self-autonomous dysregulation of host-defence, housekeeping and sensing. As a result, initiate or exaggerate proinflammatory responses and leads to neurodegeneration. We have modified the sentence to express the information better.

Thus, homeostasis in CNS microglia is necessary to disrupt neurodegenerative disease pathology and progression.

Response: In homeostatic condition, microglia plays their role as neuroprotector, while same microglia exaggerate or initiate neuroinflammation and neurotoxicity when loses their control. In this sentence we tried to mean that it is necessary to maintain the harmony of microglial activation. Uncontrolled activated microglia are the major pathogenesis of several neurodegenerative diseases.